# Using Wearable and Non-Invasive Sensors to Measure Swallowing Function: Detection, Verification, and Clinical Application

**DOI:** 10.3390/s19112624

**Published:** 2019-06-09

**Authors:** Wann-Yun Shieh, Chin-Man Wang, Hsin-Yi Kathy Cheng, Chen-Hsiang Wang

**Affiliations:** 1Department of Computer Science and Information Engineering, College of Engineering, Chang Gung University, No. 259, Wen-Hwa 1st Road, Kwei-Shan, Tao-Yuan 333, Taiwan; wyshieh@mail.cgu.edu.tw; 2Department of Physical Medicine and Rehabilitation, Chang Gung Memorial Hospital, 5 Fu-Hsing Street, Kwei-Shan, Tao-Yuan 333, Taiwan; 3Graduate Institute of Early Intervention, College of Medicine, Chang Gung University, No. 259, Wen-Hwa 1st Road, Kwei-Shan, Tao-Yuan 333, Taiwan; kcheng@mail.cgu.edu.tw; 4Graduate Institute of Medical Mechatronics, College of Engineering, Chang Gung University, No. 259, Wen-Hwa 1st Road, Kwei-Shan, Tao-Yuan 333, Taiwan; wyshieh88@gmail.com

**Keywords:** deglutition, respiration, thyroid cartilage, larynx

## Abstract

Background: A widely used method for assessing swallowing dysfunction is the videofluoroscopic swallow study (VFSS) examination. However, this method has a risk of radiation exposure. Therefore, using wearable, non-invasive and radiation-free sensors to assess swallowing function has become a research trend. This study addresses the use of a surface electromyography sensor, a nasal airflow sensor, and a force sensing resistor sensor to monitor the coordination of respiration and larynx movement which are considered the major indicators of the swallowing function. The demand for an autodetection program that identifies the swallowing patterns from multiple sensors is raised. The main goal of this study is to show that the sensor-based measurement using the proposed detection program is able to detect early-stage swallowing disorders, which specifically, are useful for the assessment of the coordination between swallowing and respiration. Methods: Three sensors were used to collect the signals from submental muscle, nasal cavity, and thyroid cartilage, respectively, during swallowing. An analytic swallowing model was proposed based on these sensors. A set of temporal parameters related to the swallowing events in this model were defined and measured by an autodetection algorithm. The verification of this algorithm was accomplished by comparing the results from the sensors with the results from the VFSS. A clinical application of the long-term smoking effect on the swallowing function was detected by the proposed sensors and the program. Results: The verification results showed that the swallowing patterns obtained from the sensors strongly correlated with the laryngeal movement monitored from the VFSS. The temporal parameters measured from these two methods had insignificant delays which were all smaller than 0.03 s. In the smoking effect application, this study showed that the differences between the swallowing function of smoking and nonsmoking participants, as well as their disorders, is revealed by the sensor-based method without the VFSS examination. Conclusions: This study showed that the sensor-based non-invasive measurement with the proposed detection algorithm is a viable method for temporal parameter measurement of the swallowing function.

## 1. Introduction

Swallowing is a process to convey the food bolus or water from the oral cavity to the pharynx and into the esophagus. Successful swallowing requires good coordination between the nasopharyngeal and oropharyngeal movements [1,2,3,4]. Many diseases, such as neurological disease, chronic indigestion disorder, gastroesophageal reflux disease, cancer, and other diseases of the head and neck, impair this coordination and cause swallowing difficulty [2,5,6]. Consequently, if swallowing dysfunction (also called dysphagia) is not assessed and treated early, many complications, such as dehydration, malnutrition, choking injuries, or aspiration pneumonia, may occur. Moreover, all of these complications lead to longer hospital stays and healthcare expenditures.

A widely used method to assess swallowing dysfunction is the videofluoroscopic swallowing study (VFSS) [7,8]. It uses X-ray photography to examine the laryngeal motion, especially the hyoid bone movement, to determine how much contrast medium, typically the barium bolus, remains in the oral cavity and pharynx during swallowing. This method is considered a gold standard because it helps the physician record the physiological movement in the larynx using high-density images (30 frames/s). However, certain risks exist for patients due to barium ingestion and radiation exposure. Moreover, for patients with poor mobility, it is not easy to conduct this examination near the bedside. There are other methods to evaluate swallowing dysfunction, for example, the use of a fiber optic endoscope to check the oropharynx and hypopharynx before and after swallowing, or the use of needle electromyography to monitor the response of the submental muscles. Both of these methods, however, are invasive measurements. 

The current trend for the bedside swallowing test is the use of radiation-free and non-invasive approaches which paste the sensors on the surface of the larynx to detect swallowing events and perform temporal measurement. Sazonov et al. [9] and Zoratto et al. [10] glued a sound sensor (microphone) over the laryngopharynx to detect ingestion behaviors, such as chewing and swallowing. Lee et al. placed an accelerometer at the midline of the anterior neck below the thyroid cartilage to measure the upward and downward motions of the larynx [11]. Li et al. used a bend sensor, which responded to a change of angles on a metal pad, to record the hyoid bone movement during swallowing [12]. Ball et al. designed an apparatus based on a three-bulb silicon array to measure the tongue pressure which affects the swallowing function in the oral stage [13]. 

Instead of using a single sensor to measure swallowing behavior, numerous studies used multiple sensors to evaluate the coordination between swallowing and respiration [14,15,16]. This coordination is crucial to the swallowing assessment because the entrance of the esophagus is in close proximity to the larynx, and both air and the swallowed bolus share a common pathway through the pharynx. Previous studies have mentioned that breathing and swallowing are physiologically linked to ensure smooth gas exchange during oronasal breathing and to prevent suffocation, aspiration pneumonia, and severe respiratory failure during swallowing [1,2]. Martin-Harris et al. used the VFSS method and a respiration recorder to measure the pharyngeal and laryngeal swallowing events [2]. Esteves et al. used a transducer to measure the hyoid-larynx complex and recorded the nasal airflow to measure the respiration during swallowing [4]. Wang et al. further proposed an integrated method which included the detection of nasal airflow, surface electromyography (sEMG) on the submental muscle, and the movement of the thyroid cartilage [15,16]. 

All of these studies have revealed an important demand for an integrated autodetection program which would analyze the swallowing patterns using sensors, and identify the timing of each pattern during swallowing. Such a program provides not only objective measurement, but also reduces the time for analysis of large amounts of data. On the basis of this framework, an autodetection program for swallowing and respiration signals was proposed. The signals came from three sensors: (1) a nasal cannula in front of the nasal cavity, (2) a pair of sEMG electrodes on the left and right side of the submental muscle, and (3) a force sensing resistor (FSR) at a position below the thyroid cartilage. The three sensors were necessary for the combinational measurement. First, since the submental muscle activity corresponded with upward laryngeal movement [15], the measurement of submental sEMG increased the reliability and accuracy for detecting the appearance of the targeted swallowing duration. Second, the measurement of nasal airflow reflected the respiratory control and airway apnea during water swallowing. Third, the thyroid cartilage movement ensured that the water or bolus was pushed down smoothly without leakage to the trachea. Therefore, if the signals from the three sensors were time-locked on a frame, the coordination of swallowing and respiration was evaluated simultaneously. This could help physicians obtain the combinational analysis results accurately and efficiently in a non-invasive manner. To meet this requirement, an algorithm was developed to scan the signals and report the temporal parameters among the swallowing events from the sensors. The main goal of this study is to show that the sensor-based measurement with the proposed detection algorithm is able to detect early-stage swallowing disorders, and, specifically, is useful for the assessment of the coordination between swallowing and respiration.

To verify this study, the temporal parameters from the detection of the sensors were compared with the physiological movement of the thyroid cartilage recorded by the VFSS. In order to show that sensor-based measurement using an autodetection program is a viable method, a clinical application was applied to the long-term smoking effect on the swallowing function. According to previous studies, smoke, which has a high air temperature and harmful substances, hurts the oral and oropharyngeal mucosa which compromise the pharyngeal functions of nerve endings and reflexive pharyngeal swallow, and consequently, lead to an impairment of the swallowing functions [17,18]. This study also showed that the proposed sensors could identify the differences between the swallowing function of smokers and nonsmokers.

## 2. Method

### 2.1. Materials and Signal Analysis

#### 2.1.1. Sensors

Figure 1 shows the FSR used to measure the thyroid cartilage movement. It is similar to piezoresistive sensors where resistance will change proportional to the force pressed on the surface. The thinness of the FSR is less than 1 mm which is fixed on the surface of the throat by an elastic belt with the least interference in swallowing. 

A pair of Biopac bipolar electrodes (Biopac EL500, BIOPAC systems Inc., Goleta, CA, USA) were adhered around the center of the surface of the submental muscle in a bipolar configuration to detect contractions during swallowing. The Biopac bipolar electrodes provided a differential EMG measurement on the same area of the muscles. When applying the electrodes, the area of skin was cleaned with an alcohol wipe. The EMG signal was sampled at 1024 Hz and amplified by a factor of 1000 with a differential amplifier and it was transmitted through a wireless EMG transmitter (Wireless Dynamometry-EMG BioNomadix Transmitter, BIOPAC systems Inc., Goleta, CA, USA). The raw data from the EMG channel were primarily bandlimited from 5.0 Hz to 500 Hz, with subfiltering options. 

A nasal airflow cannula was placed in front of the nasal cavity to sense respiration. Figure 2 illustrates how a participant wore the sensors in the test. All signals (i.e., FSR, sEMG, and nasal airflow) were synchronized and recorded by a data collector (BIOPAC MP100, BIOPAC systems Inc., Goleta, CA, USA) and the waveforms were presented by the original signal amplitude in the voltage domain (Volts) without translation.

#### 2.1.2. Analysis of Sensor Signals

Figure 3 shows the synchronized signals of the sensors from a healthy participant swallowing 10 mL of room temperature water. The sEMG waveform shows that the submental muscles raised a positive response during (E3–E1) with the largest force at E2. At the same time, the nasal airflow waveform shows a corresponding respiration pause (A2–A1), which is a necessary protective phenomenon to allow for safe swallowing without aspiration. In this study, this apnea was measured using the nasal airflow, instead of the trachea airflow because the trachea airflow measurement needed deeper and invasive sensing which was an additional risk (choking or infection) of the test and made the measurement uncomfortable. The FSR waveform typically shows a W-shaped response which represents two phases of the thyroid cartilage excursion. The first phase (F2–F1) represents the movement of the thyroid cartilage, upward and forward, blocking the trachea to ensure that the water successfully passes the pharynx to the esophagus. The second phase (F3–F2) represents the recovery of the thyroid cartilage back to the original position. Table 1 shows the parameters defined for describing the model of Figure 3.

The coordination of respiration and larynx movement plays an important role in safe swallowing, and it involves time-related events that are controlled by the central integration of swallowing and airway reflexes in the brain stem [19]. Therefore, the parameters in Table 1 were used to measure the latency and duration among those time-related events. Another analysis of the coordination between swallowing and respiration is the respiratory phase patterns (RPPs), which involved the expiration (EX) or inspiration (IN) patterns pre- and post-swallowing [19], for example, the nasal airflow signal shown in Figure 3. The RPP of Figure 3 was “EX/IN” because the positive pulse before A1 (pre-swallowing) was an expiration and the negative pulse after A2 (post-swallowing) was an inspiration. Any subject could have one of four possible RPPs (EX/EX, EX/IN, IN/EX, and IN/IN). As the EX/EX pattern was considered the major physiologically protective type [19,20], the other three minor patterns were grouped into a non-EX/EX pattern. The ratios of both patterns (EX/EX and non-EX/EX) were also investigated in the results.

#### 2.1.3. Autodetection Program

Assume that the waveform of each signal in Figure 3 is expressed as a function, $F_{sgl}\left(t\right)$, where sgl
∈ {sEMG, Nasal airflow, FSR}, and 0≤
*t*
≤
*T (T*, the end time of the measurement). Figure 4 summarizes the six possible deflection patterns in the waveforms which can be used to identify the onset or the offset for each signal. To detect each deflection, a voltage baseline was fixed at a certain voltage level for reference. This level was obtained by asking a participant to sit stably without oral movements for a short time before the swallowing. The baseline of each signal, thus, is defined as the mean of the signal from the first few seconds (called $M_{sgl}$). Assuming the standard deviation of the signal in this period is $SD_{sgl}$, two thresholds around the baseline are defined as:(1)$$Thr_{1}\;=\;M_{sgl}+\;\alpha \;\cdot \;SD_{sgl}$$
(2)$${\mathrm{Thr}}_{2}\;=\;M_{\mathrm{sgl}}\;–\;\beta \;\cdot {\mathrm{SD}}_{\mathrm{sgl}}$$
where, α and β are two constants to adjust the level of the thresholds. (In the following experiment, α = β = 2, based on experimental trials). By comparing each point of the signal with the thresholds ($Thr_{1}$ or $Thr_{2}$), we detected when the waveform changed upward, or downward. We defined an indicator function $D_{sgl}\left(t\right)$ to save the comparing results, where
(3)$$D_{sgl}\left(t\right)\;=\{\begin{array}{c} \;1,\;if\;F_{sgl}\left(t\right)\;>\;Thr_{1},\;\\ \;-1,\;if\;F_{sgl}\left(t\right)\;<\;Thr_{2}\\ 0,\;otherwise. \end{array},$$

Algorithm 1 shows the algorithms to detect the onset and offset for sEMG, nasal airflow, and FSR waveform, respectively, by counting $D_{sgl}\left(t\right)$. A sliding window of the size *N* points was initiated to scan along the signal. When the sliding window moves, the values of $D_{sgl}\left(t\right)$ in the window are summed up. Since sEMG and FSR waveforms are purely positive and negative responses as compared with their baselines, the onset of both signals is marked by detecting the first appearance of Pattern 1 and Pattern 3, and the offsets are marked by the appearance of Pattern 2 and Pattern 4 in Figure 4. For the nasal airflow waveform, different from the other two signals, the onset of apnea will appear when the sum of $D_{sgl}\left(t\right)$ first descends to a small range around the baseline, and the offset appears when the sum of $D_{sgl}\left(t\right)\;$starts to increase or decrease away from the baseline. Finally, the turn-around points in the FSR waveform are detected by looking for the local maximal (Pattern 5) or minimal points (Pattern 6) between the onset and the offset.

**Algorithm 1:** The detection of the onset and the offset for sEMG, nasal airflow, and FSR.

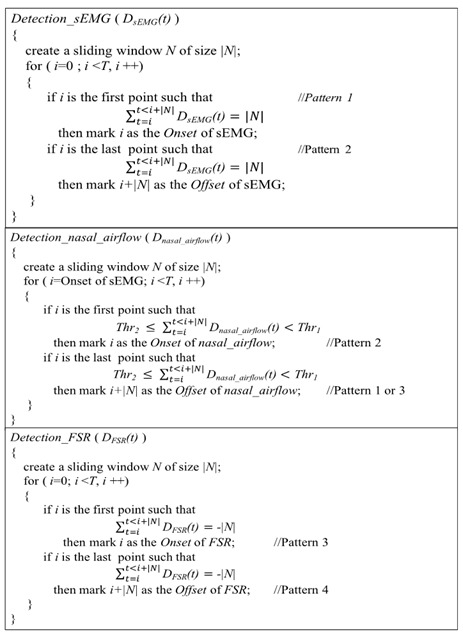



Figure 5 shows the LabVIEW [21] program we developed according to the Algorithm 1. The windows on the left side of Figure 5 show the waveforms with each onset, offset, and turn-around points labeled by the cursors and the positions of each cursor, as well as the parameters defined in Table 1, are listed on the right side. The timing of the parameters was exported to the files for statistics. 

### 2.2. Participants: Smoking and Nonsmoking Groups

We recruited 45 male participants, aged 30–50 years, to participate in the test (Table 2). They were divided into two groups: the first group consisted of 26 nonsmoking participants and the second group consisted of 19 smoking participants (Table 2). All participants in the second group had smoked more than 10 cigarettes daily for over 10 years. Exclusion criteria included any past history or symptoms of dysphagia, neurological disease, cardiopulmonary disease, indigestion disorders, gastroesophageal reflux disease, cancer or other diseases of the head and neck, and the current use of medications with known effects on swallowing or breathing. Prior to acceptance into the study, all participants underwent a standardized oral mechanism examination, and a water swallow screening test was conducted by a registered speech language pathologist to confirm the absence of any clinical signs of dysphagia. Each participant signed informed consent prior to the test. Ethics approval was granted by the Chang Gung Memorial Hospital.

Each participant was asked to swallow five volumes of room temperature water: 1, 3, 5, 10, and 20 mL. Each volume of the test needed to be repeated for three trials and recorded individually. Between any two volumes of tests, the participants rested for 3 min. Participants were reminded to swallow normally within their ability. For safety, we asked the participants to start with the smallest volume of water and then increase the volume sequentially. The maximum swallowing volume was limited at 20 mL according to the previous studies [15,16]. 

The analyses were performed using the SPSS software version 12.0 (SPSS Inc., Chicago, IL, USA) [22]. The data obtained for the three swallowing trials of the same volume were averaged. The independent two-sample t-test was used for parametric testing to examine and compare the differences between two groups. The difference between two groups with *p* < 0.05 was considered statistically significant. 

### 2.3. Verification of the Method

The program shown in Algorithm 1 was verified by comparing the parameters measured by the sensors (in Figure 5) with the thyroid cartilage movement recorded by the VFSS (Siemens Luminos, 30 frames/s). The frame rate of the VFSS instrument was 30 frames/s. The recorded video was split into individual frames and we asked a physician and a therapist to double check the timing of the thyroid cartilage movement. Six volunteers in total wore the sensors and participated in the VFSS test. Every participant swallowed 10 mL of 30% (w/v) barium bolus, and this was repeated 5 times. In each VFSS video, three sequential events were monitored, and their timing was recorded as a reference for describing the state of the laryngeal movements (Table 3 and Figure 6). Accordingly, three intervals (TC2–TC1, TC3–TC2, and T3–T1) obtained from the VFSS were compared with the intervals detected from the sensor waveforms (F2–F1, F3–F2, and F3–F1). The apnea time (SAD) and the sEMG duration, sEMGD, were not compared because the respiration events and the submental muscle movement could not be monitored visually by the VFSS. 

## 3. Results

### 3.1. Verification of the Algorithm

Table 4 shows that the means of difference between each interval from FSR (F1–F2, F2–F3, and F1–F3) and each corresponding interval from VFSS (TC1–TC2, TC2–TC3, and TC1–TC3) were all smaller than 0.02 sec (i.e., diff ratio < 3%, *p* > 0.05 by the dependent t-test). Here the dependent t-test was adopted because two measurements were performed on the same participants. Table 5 further shows that the timing difference between the starting events of each interval: F1 and TC1, F2 and TC2, and F3 and TC3, were all smaller than 0.03 sec. Since the detection of F1, F2, and F3 depended on the threshold setting (Equation 1 and Equation 2) in the proposed algorithm, the time points of TC1, TC2, and TC3 were all faster than F1, F2, and F3, respectively. 

### 3.2. Clinical Measurement on Smoking and Nonsmoking Participants

#### 3.2.1. Submental Muscle Results: sEMGD

Figure 7 shows the average sEMGD between the nonsmoking and smoking groups. The piecemeal swallowing (also known as “piecemeal deglutition” [23]), where the participant took multiple swallows to ingest the full amount of water, were excluded because its waveform did not fit the normal swallowing model in Figure 3. The results showed that the smoking group had a longer sEMGD for each volume of water than the nonsmoking group, which means that their hyoid bone tends to have a longer time to push the water backward during swallowing. It is noticeable that the shortest sEMGD happened at 5 mL, instead of 1 or 3 mL. This is reasonable because most participants tended to use a larger force within a longer time to swallow a minute volume of water. Nevertheless, the smoking participants took even longer. The t-test results proved this because even with 3 mL of water the smoking participants showed a significant difference as compared with the nonsmoking participants (*p* = 0.04).

#### 3.2.2. Nasal Airflow Results: SAD

Figure 8 shows that the apnea time (SAD) of both groups increased along with increasing volumes of water, but the smoking group had a slightly higher increase rate. Moreover, the t-test results revealed that both groups had very obvious differences in each volume of water (*p* = 0.032, 0.047, 0.032, 0.001, 0.009 from 1 mL to 20 mL). For the respiratory phase patterns pre- and post-swallowing, there were no significant differences in the number of swallows in pre- and post-swallowing respiratory phase patterns between the two groups (EX-EX/non-EX-EX swallows of smoking vs. EX-EX/non-EX-EX swallows of nonsmoking from 1 mL to 20 mL: 12/66 vs. 8/49, 11/67 vs. 8/49, 14/64 vs. 10/47, 8/70 vs. 7/50, and 8/70 vs. 7/50). 

#### 3.2.3. FSR Results: Onset Latency (OL), Total Excursion Time (TET), Excursion Time (ET), Duration of Second Deflection (DEFD)

Figure 9 shows the results of the thyroid cartilage movement from the FSR measurement. In Figure 9a, the negative OL from 3 mL to 20 mL shows that the participants had an earlier onset of the thyroid cartilage movement than the late onset of submental sEMG. The smoking group showed shorter latencies of sEMG on average, except the 5 mL test, but with no significant difference as compared with the nonsmoking group. Figure 9b shows that the smoking group needed longer TET to complete the whole swallowing in all tests. Moreover, the differences between the two groups were very obvious (*p* ~ 0.000 for 1 mL, and 5 mL; *p* = 0.001, 0.019, 0.038 for 10 mL to 20 mL) in the TET comparison. To investigate which part caused the differences, we compared the first phase (ET) and the second phase (DEFD) of the thyroid cartilage movement separately for both groups. Figure 9c,d show that the smoking participants performed almost equal in the ET (*p* = 0.217, 0.137, 0.636, 0.720, 0.750 from 1 mL to 20 mL) but much longer in the DEFD (*p* ~ 0.000 for 1 mL, and 5 mL; *p* = 0.001, 0.004, 0.029 from 10 mL to 20 mL) as compared with the nonsmoking participants. 

#### 3.2.4. Piecemeal Swallowing

When the water volume exceeds a limit (typically 20 mL), numerous participants resort to piecemeal swallowing. This involves dividing the bolus into smaller pieces and swallowing in several gulps. A swallowing limit below 20 mL was considered inconspicuous dysphagia in neurogenic disorders [23,24].

Piecemeal swallowing is detected easily by monitoring the FSR waveform. Figure 10 shows one case, coming from a smoking participant (male, aged 34 years) in a 20 mL test. The FSR waveform shows that the piecemeal swallowing causes a series of pulses that appear sequentially right after the first swallow.

Table 6 shows the percentages of the piecemeal swallowing over all the trials for the nonsmoking and smoking groups in the 5 mL to 20 mL tests. (In the 1 mL and 3 mL tests, piecemeal swallowing did not happen in either group because of the minute volume.) In the 5 mL test, the percentage of piecemeal swallowing for the smoking group exceeded 10%. When the volume was increased to 20 mL, the percentage increased to 46%, as compared with 29% for the nonsmoking group. 

## 4. Discussion

Swallowing disorders cause many consequences [5,6,7,23]. Understanding the coordination between swallowing and respiration is crucial for health assessment. More and more studies have used non-invasive sensors in the early stage to investigate swallowing disorders while circumventing the problems of the invasive methods. The sonic method [9] detected the gulp sounds for ingestion behavior investigation but it has the problem of noise intervention. The motion sensor [11] measured the upward and downward motions of the larynx but it could fail if the participant’s head or torso moves. The bend sensor [12] does not have those intervention problems, but it does not exactly fit the throat surface of each participant. This study, instead, used the FSR sensor to measure the thyroid cartilage movement which had high sensitivity and small size advantages. This study, however, did not measure the activation of thyroid muscles. In normal swallowing, submental muscles and thyroid muscles contract to pull the hyoid bone and thyroid cartilage anteriorly [24]. The correlation between the thyroid cartilage movement and the activation of thyroid muscles is an important issue for assessment of the swallowing function, which we have considered for future work. In addition, respiration was recorded by monitoring the nasal airflow, not the tracheal airflow. The tracheal airflow is more suitable for reflecting the clearance of trachea during swallowing. The difference and the correlation between the nasal airflow and the tracheal airflow are compared simultaneously by checking the airflow in the tracheostomized patients without plugging and with plugging. 

The major difference between this study and previous ones is that an autodetection program was proposed to identify the swallowing events among the FSR, submental sEMG, and nasal airflow signals. It is helpful to researchers for measuring the parameters of swallowing and respiration, as well as their correlations, in a faster way. Moreover, the comparison of this program with the VFSS showed that the physiological movement of the laryngeal can be described objectively through non-invasive sensors. The verification results showed that the non-invasive FSR sensor could be considered a reliable way to measure the physiological laryngeal movement during swallowing.

An application for testing the effect of smoking was included in this study. The FSR measurements showed that the smoking participants on average had a longer thyroid cartilage movement time (TET) and returning time (DEFD). This means that the smoking participants took longer time to return the thyroid cartilage back to the original position after swallowing. This symptom typically appears at the early stage of dysphagia [23]. A significant difference also appeared in the submental sEMG duration time (sEMGD) and the swallowing apnea time (SAD), as compared with nonsmoking participants. Other previous studies have revealed a similar effect that smoking will hurt oral and pharyngeal mucous as well as the sensory receptors [17,18]. This study further proves that smoking could affect the participant’s respiration by a prolonged apnea time for safe swallowing. 

Few studies have addressed the issue of piecemeal swallowing for people who smoke. This study showed that the smoking group had a highly significant increase in the percentage of piecemeal swallowing, as compared with the nonsmoking group. Piecemeal swallowing is a protective phenomenon which scarifies efficiency to prevent participants from choking when the swallowing volume exceeds a person’s limit. Previous studies have shown that frequent piecemeal swallowing demonstrates a high risk of dysphagia in neurological diseases [23,25,26]. This work further revealed that people who smoke appeared to have a greater chance of this phenomena.

In this study, three sensors were adapted for the combinational measurement. Each sensor was attached on the surface of the skin and fixed by wires or belts. Nevertheless, the setup time was still longer than the single sensor measurement method. However, the single sensor measurement does not reveal the overall swallowing function from different physiological reactions during swallowing. While this study focused on the three sensors to detect the coordination of swallowing and respiration, other sensors could be integrated with this study to provide a more complete measurement. For example, the researchers of [13] have shown that the tongue pressure in the anterior oral stage could also be a very important factor in swallowing dysfunction. The integration of the tongue pressure measurement with this study has become an open issue and is under investigation. 

## 5. Conclusions

In this study, a non-invasive, sensor-based approach to analyze the coordination between respiration and swallowing was proposed. The testing results show that this approach is able to differentiate the swallowing patterns, with almost no significant delays as compared with the VFSS. The proposed autodetection program is useful for identifying and recording swallowing events in a fast and convenient way. For future work, the approach of this study could be applied to other swallowing disorders, such as in patients with stroke or Parkinson disease, to justify if this approach is adaptable for different requirements. 

## Figures and Tables

**Figure 1 sensors-19-02624-f001:**
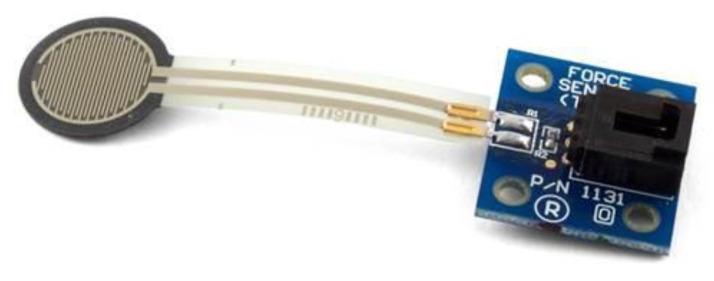
Force sensing resistor (FSR) sensor.

**Figure 2 sensors-19-02624-f002:**
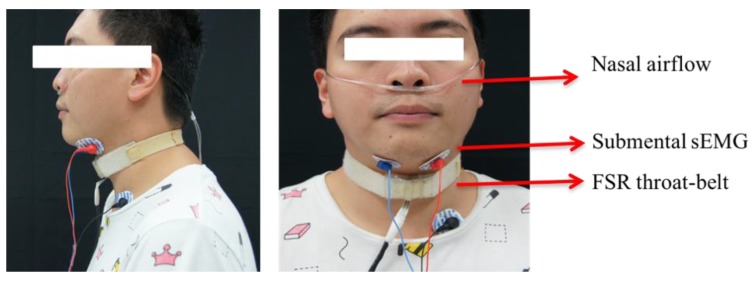
The usage of FSR, submental surface electromyography (sEMG), and nasal airflow cannula.

**Figure 3 sensors-19-02624-f003:**
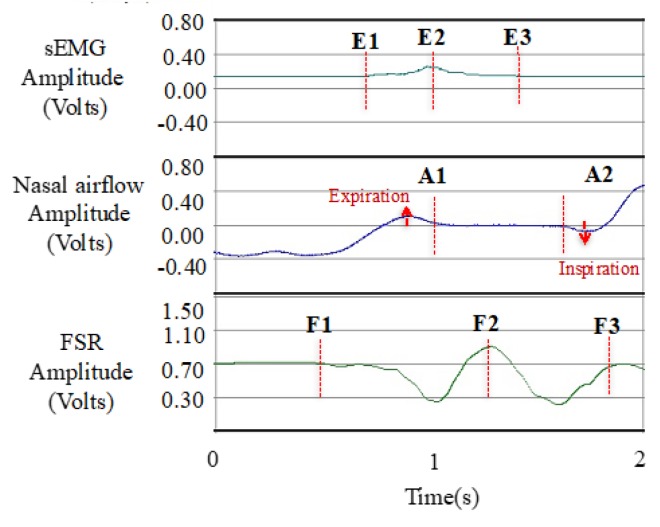
Swallowing and respiration signals from the sensors. (E3–E1): the duration of the submental muscle response, (A2–A1): the duration of the respiration pause, (F2–F1): the duration of the thyroid cartilage moving upward and forward to block the trachea, (F3–F2): the duration of the thyroid cartilage returning back to the original position.

**Figure 4 sensors-19-02624-f004:**
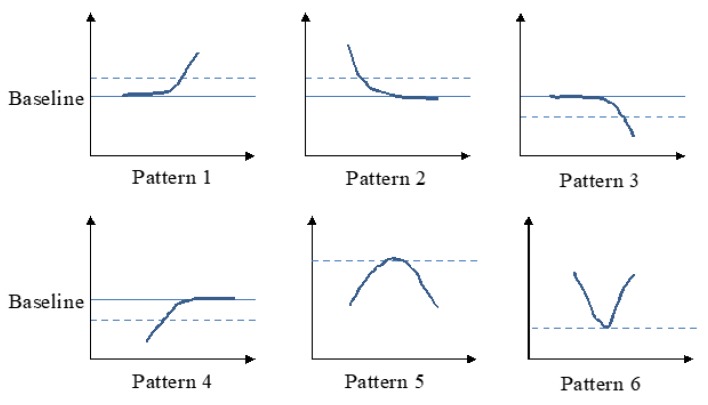
Six patterns which are found as the features of the onset or offset points in Figure 3.

**Figure 5 sensors-19-02624-f005:**
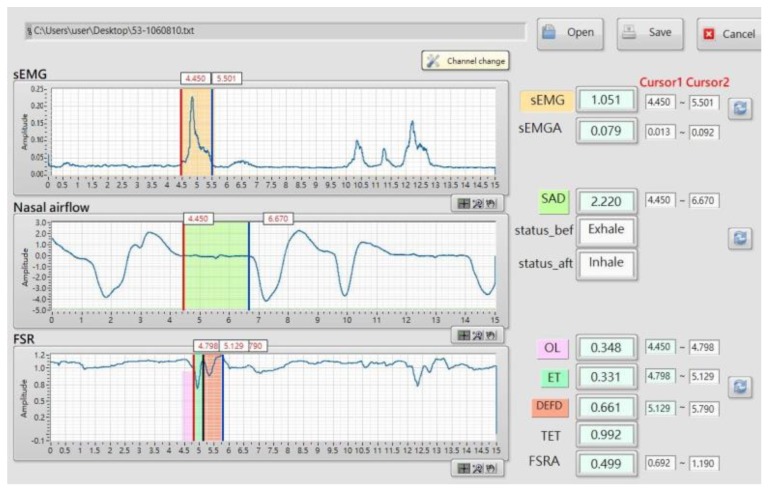
The program we developed from the algorithm in Algorithm 1.

**Figure 6 sensors-19-02624-f006:**
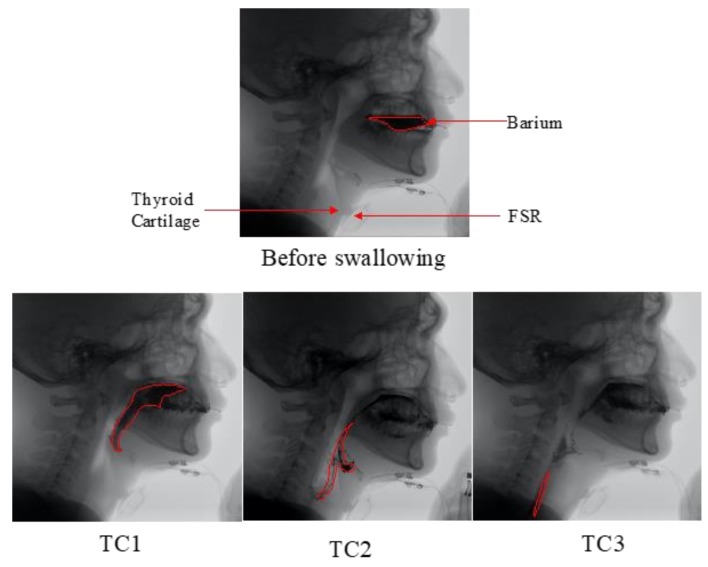
Sequential VFSS captures: Before swallowing, TC1, TC2, and TC3. The positions of the thyroid cartilage and the FSR were labeled in the first capture.

**Figure 7 sensors-19-02624-f007:**
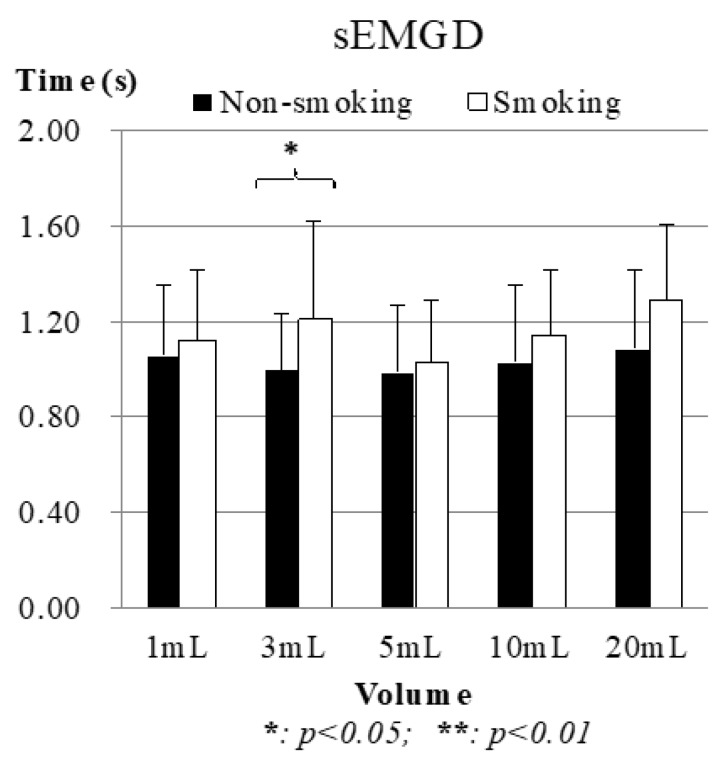
Average results of surface electromyography duration (sEMGD). Piecemeal swallowing cases were excluded.

**Figure 8 sensors-19-02624-f008:**
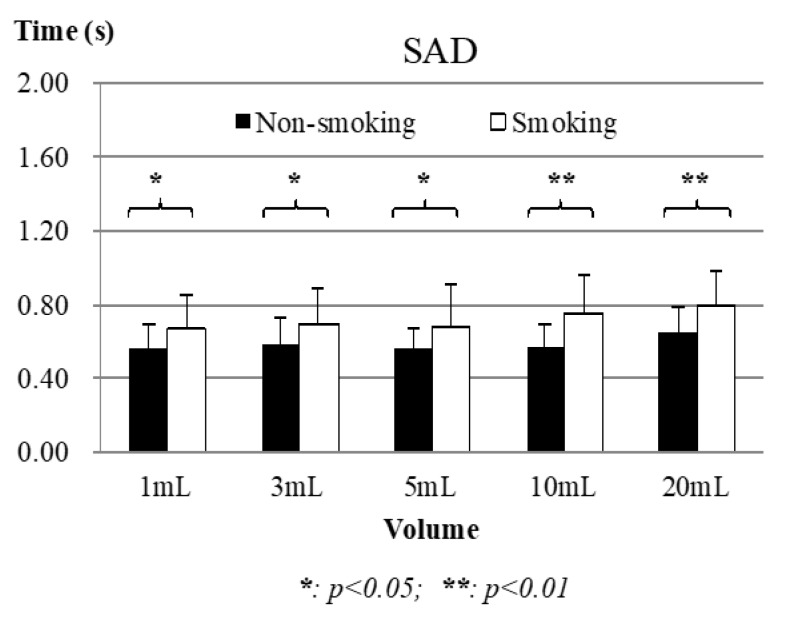
Apnea time (SAD) comparison results between nonsmoking and smoking groups. Piecemeal swallowing cases were excluded.

**Figure 9 sensors-19-02624-f009:**
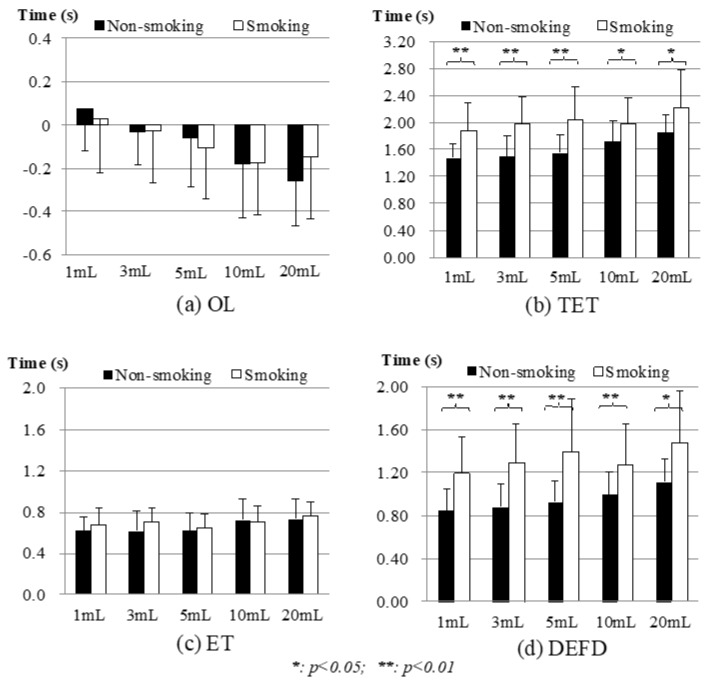
Comparison of FSR signals between nonsmoking and smoking groups: (**a**) Onset Latency (OL); (**b**) Total Excursion Time (TET); (**c**) Excursion Time (ET); (**d**) Duration of Second Deflection (DEFD). Piecemeal swallowing cases were excluded.

**Figure 10 sensors-19-02624-f010:**
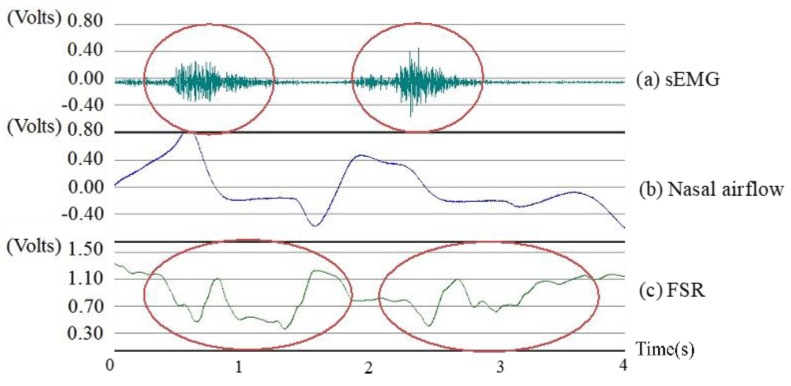
FSR waveform of a case of piecemeal swallowing for 20 mL of water.

**Table 1 sensors-19-02624-t001:** Parameters of the respiration and swallowing signals.

Parameter	Definition	Calculation
sEMGD	sEMG duration	(E3–E1)
SAD	swallowing apnea duration	(A2–A1)
OL	onset latency between sEMG and the thyroid cartilage	(F1–E1)
ET	excursion time of the first phase in the W-shaped signal of the thyroid cartilage	(F2–F1)
DEFD	duration of second deflection in the W-shaped response of the thyroid cartilage	(F3–F2)
TET	total excursion time of the thyroid cartilage excursion	(F3–F1)

**Table 2 sensors-19-02624-t002:** Participants in the nonsmoking and smoking groups.

Group	Nonsmoking	Smoking
Number of participants	26	19
Average age	38.12 ± 6.45 (years)	37.68 ± 7.13 (years)
Average smoking period	0	16.21 ± 6.04 (years)
Number of cigarettes daily	0	17.95 ± 7.26

**Table 3 sensors-19-02624-t003:** Events which were monitored using the videofluoroscopic swallow study (VFSS).

VFSS Event	Movement of the Thyroid Cartilage
TC1	The thyroid cartilage started to move upward
TC2	The thyroid cartilage moved toward the front position
TC3	The thyroid cartilage returned to the original position

**Table 4 sensors-19-02624-t004:** Comparison of the VFSS intervals (sec) with the FSR intervals (sec).

VFSS Interval	Mean ± SD	FSR Interval	Mean ± SD	Mean of Diff	Diff Ratio	Correl	*p*
(TC1–TC2)	0.782 ± 0.02	F1–F2	0.762 ± 0.32	0.020	2.7%	0.91	0.84
(TC2–TC3)	0.580 ± 0.13	F2–F3	0.591 ± 0.23	0.011	1.9%	0.92	0.75
(TC1–TC3)	1.363 ± 0.16	F1–F3	1.353 ± 0.35	0.010	0.8%	0.96	0.53

(SD, standard deviation; Diff =|VFSS interval – FSR interval|; Diff ratio = Diff/VFSS interval; Correl, correlation; *p* < 0.05 difference is statistically significant).

**Table 5 sensors-19-02624-t005:** The timing difference between the FSR and the thyroid cartilage movement using the VFSS.

Timing Events (FSR, VFSS)	Mean of Timing Difference (s) ± SD
(F1, TC1)	0.021 ± 0.01
(F2, TC2)	0.013 ± 0.02
(F3, TC3)	0.010 ± 0.01

**Table 6 sensors-19-02624-t006:** Frequency of piecemeal swallowing.

	Nonsmoking	Smoking	*p*-Value
Number of subjects	26	19	-
Total trials in each group	78	57	-
Piecemeal swallowing of 5 mL	2 (3%)	7 (12%)	0.036 (#)
Piecemeal swallowing of 10 mL	5 (6%)	11 (19%)	0.030 (*)
Piecemeal swallowing of 15 mL	23 (29%)	26 (46%)	0.070

* represent statistically significant difference (*p* < 0.05); # the sample size is too small to claim difference.

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
