# Peer review of "Using Wearable and Non-Invasive Sensors to Measure Swallowing Function: Detection, Verification, and Clinical Application"

_sensors, 2019, doi:10.3390/s19112624_

Round 1

Reviewer 1 Report

The authors responded to all my comments appropriately, and I do not have any further comments.

Author Response

Thank you very much for your kindness and helpful suggestions. 

Reviewer 2 Report

Specific comments)

1)     The main goal of this study was to show that the sensor-based method was comparable with the VFSS method in measuring those timing related parameters of swallowing function. However, the results and conclusion are beyond the purpose. The manuscript is needed to be organized and focused on the main purpose.

2)     The authors are necessary to reorganize the abstract. The purpose of the study was to show that the sensor-based method is comparable with the VFSS method in measuring the timing related parameters of swallowing function. But the results described the differences of three sensors between smoker and non-smoker and concluded that the sensor-based measurement with the auto-detection program is a viable method for earlier stage swallowing function measurement.

3)     Keywords: change to MeSH term, Suggested keywords are arbitrary.

4)     Though the authors combined three sensors (breathing sensor, EMG sensor, force sensing resistor), they did not differentiate the advantage of the simultaneous measurement of three sensors.

5)     In the introduction, the authors regarded VFSS as a gold standard evaluation of the procedure. But VFSS measures the temporal movement of thyroid cartilage or hyoid bone. This motion is induced by the contractions of the related muscles. In this area, therefore, the gold standard can be changed to other measures which are more accurate and faster than VFSS. VFSS is a gold standard evaluation for detecting aspiration in clinical purpose.

6)     In line 133, the authors measured the interval between A1 and A2. In swallowing, the direction of airflow is important in the trachea. What do you think the correlation between the airflow in the trachea and nasal flow? And the time interval between the trachea and nasal cavity?

7)     Line 133. The authors set A1 and A2 of nasal airflow. In swallowing, the direction of nasal airflow is important. What do you think of setting the variable at 0.00 or turning point of the curve?

8)     The authors described the variables of three sensors compared to VFSS. As the relation between sensor based measure and VFSS, the accurate time interval and correlation (the time interval, specific feature, reliability and delay time) between the sensors and VFSS are important and necessary to be presented.

9)     Line 215, SPSS version?

10)  Some sentences should be moved from the results section to the method section or discussion section.

11)  Line 285-286: This sentence is necessary to move to the discussion section. The contents of the discussion are relatively short and poor.

12)  Methods section can be divided 1) the explanation of the equipment (three sensors), 2) the correlation and verification of this equipment with VFSS and 3) the comparison between smoker and non-smoker.

13)  The results section should be started on line 231.

14)  Line 235: The time interval between F1 and TC1, F2 and TC2, and F3 and TC3 were less than 0.03 sec. Which parameters were faster between F and TC?

15)  In a kinematic analysis of VFSS, we can easily find a pres-wallow movement of hyoid bone and thyroid cartilage. How the authors managed these movements in VFSS?

Author Response

Paper ID:sensors-509394

Paper Title: Using Wearable and Non-invasive Sensors to Measure Swallowing    

          Function: Detection, Verification, and Clinical Application

Dear Editor and Reviewers,

    We have carefully revised the manuscript according to your helpful comments. The major revisions in this version included:

(1) the abstract has been reorganized and the description of the main goal has been modified,

(2) the reasons of the combinational measurement of the three sensors have been added,

(3) Section 2 has been reorganized into three sections (method, participants, and verification),

(4) the discussions of the results in Section 3 have been moved to Section 4, and  

(5) Section 5 Conclusion has been modified.

We are looking forward to your suggestions. 

Best Regards,

Wann-Yun

Reviewer 3 Report

The manuscript, “Using Wearable and Non-invasive Sensors to Measure Swallowing Function: Detection, Verification, and Clinical Application” by Shieh et al., reports measurements of swallowing function from 45 male population with (n = 19) or without (n = 26) heavy smoking experiences.  The swallowing function was measured from three sensors: (1) nasal cannula, (2) surface electromyogram (sEMG) electrodes on submental muscles, and (3) force sensing resistor (FSR) from below the thyroid cartilage. All these sensors are commercially available ones, so hardware itself is not a novelty in this study. A labview program was used to detect swallowing and respiration signals automatically from three different cues.

The data obtained in this study may add the dataset that can be used by research community.  These results are worth publishing for future studies.  However, the reviewer feels that a few crucial components are missing in the current manuscript to be an archival research paper.  There are a few errors that may be addressed as well.

(1)   There has been numerous studies that used multiple sensors to detect the details of swallowing event.  From a wearer’s (and a consumer’s) perspective, simplest a single functional sensor is the most effective if the sensor provides useful-enough information.  Multiple sensor studies, however, kept adding multiple functionalities to add more useful information.  In this study, discussion is missing why the combination of three sensors (cannula, sEMG, and FSR) should be used.  In addition, the discussion part must contain contents that describe indispensable reasons that tell how the combination of the three provides unique information to study swallowing events.  Current paper reads as if the results from three sensors were described in a simply parallel manner.  In short, what is the synergy?  Tell the readers from theoretical and result perspectives.  For this reviewer, this ‘combination’ is the only claimable novelty of this study, so justifying this combination is extremely important to assess the publication of this study.

(2)   In Line 166 (algorithm), how (alpha) = (beta) = 2 is optimal?

(3)   Line 236 – 238, please provide a citation about the ‘other previous studies’.

(4)   Sections 3.2.2 and 3.2.3 (both):  Why was the piecemeal swallowing results excluded in the two sections?

(5)   Line 275 – 277: The authors claim that the smoking group showed shorter latencies.  The results in Figure 10(a) seems to tell the readers exactly otherwise.  Please explain.

(6)   Comparion between Fig 3 & Fig 10a. In Fig. 3, the data is for 10ml water swallowing. F1-E1 (OL) is a positive number. However, in Fig 10a, the results are negative. Please explain.

(7)   Table 6, Piecemeal swallowing in 5 mL, the sample size is too small to claim a meaningful p-value.

(8)   Lines 346 – 347 (Conclusion): “It would be very easy to apply these approaches to other swallowing disorder assessments, …”:  Why and how?  This statement is very abrupt and lacks a ground.  This problem is associated with my point (1).  The novelty of this study entirely relies on the data acquisition by three sensors.  When the combination is not justified, how can one be convinced that the approach is ‘very easy’ and useful for other disorders?  Each disorder require different necessities.  What are the real problems that this ‘combination’ can solve?

(9)   MINOR ERRORS:

-        Fig 3: What is x-axis?

-        I cannot find 4 RPPs in Fig 3.  Please mark clearly.

-        Fig 6: sEMG y-axis

-        Line 216:  What kind of t-test was used?

-        Table 5: Provide Standard Deviation

Author Response

(The authors gave the same response as above.)

Round 2

Reviewer 2 Report

1. What is the mechanism to activate FSR. According to your manuscript, it is a kind of piezoresistive sensors whose resistance will change proportional to the force pressed on the surface and used to measure the thyroid cartilage excursion. The authors intended to measure the movement of the thyroid cartilage.

Is FSR detect the activation of thyroid muscles? I am not sure whether FSR detects the movement of thyroid cartilage or the contraction of thyrohyoid muscle. This article (J Electromyogr Kinesiol. 2017 Oct;36:81-89) will be helpful to analyze the FSR. This article measured HRM, needle EMG, and VFSS simultaneously. Thyrohyoid muscle is placed at FSR sensor location. In normal swallowing, submental muscle pulls hyoid bone anteriorly and thyroid muscle’s contraction which pulls larynx anterior start after 150ms later. Then other infrahyoid muscles’ contraction starts after 3000ms later.

2. You answered well regarding the airflow in the nasal flow.

In my opinion, tracheal airflow is most suitable for analysis. You can check the tracheal airflow in tracheostomized patients without plugging and with plugging and then can compare the difference the tracheal airflow and nasal airflow to find out the difference or correlation. If you cannot address this issue, you can describe in the limitation section.

Author Response

    Thanks again for these valuable comments. We have revised the manuscript accordingly. The revisions in this version included:

(1) We add more descriptions about the issue of using FSR and nasal airflow in the discussion section (Line 350-359),

(2) We add one article in the reference list (Line 478).

We are looking forward to your suggestions. 

Best Regards,

Wann-Yun Shieh

Department of Computer Science and Information Engineering,

Chang Gung University

Reviewer 3 Report

The authors made adequate changes.

Author Response

Dear the reviewer, 

     Thank you very much for your reviewing comments and great helps.

Best Regards,

Wann-Yun

This manuscript is a resubmission of an earlier submission. The following is a list of the peer review reports and author responses from that submission.

Round 1

Reviewer 1 Report

1. The authors mention in ‘Conclusions’ that they analyzed the coordination between respiration and swallowing. However, only data presented relating to the coordination is SAD. Additional analyses are required to elucidate changes in the coordination between respiration and swallowing by smoking.

2. The authors concluded that the present approach could differentiate the swallowing patterns with almost no significant delays. However, what the authors presented is the comparison between the duration of VFSS parameters and those measured by the proposed device. There is no description regarding the delay between VFSS measures and those of the proposed devise. The authors should analyze the timing difference between E1 and HB1, F1 and TC1, and F2 and TC2.

2. The authors assert that the verification of the auto-detection program was conducted in two ways, i.e. comparison with VFSS data and comparison between non-smokers and smokers. However, the latter is not a verification but an application. Verification should include measurements under noisy environments, e.g., contamination of speech, cough, nod, or stiffened neck muscles, to test whether the proposed algorithm is still valid in the presence of such contamination.   

Author Response

Dear Reviewers,

    We have carefully revised the manuscript according to the reviewer’s comments. The major revisions include:

1.     The main goal of this study was revised in Abstract and Section 1. 

2.     Two references about the effect of smoking were added in the reference list and introduced in Section 1.

3.     The detailed sEMG configuration and connection was added in Section 2.1. 

4.     Additional analysis about the coordination of swallowing and respiration was added in Section 2.2 and Section 3.2.2.

5.     The method of Statistical Analysis was added in Section 2.6.

6.     The verification of the algorithm was revised and rewritten in Section 3.1

Other ambiguous parts were rewritten and new paragraphs and sentences were added when necessary. We provide the detailed reply to the comments in the attached file. 

We greatly appreciate your careful and expeditious review of this manuscript.

Kind Regards,

Wann-Yun

Reviewer 2 Report

The purpose of the study is not clear. Is this a viability study? It is not clear why you decided to include a comparisson between smokers and non-smokers.

This study does not verify the method or the algorigthm. Time-period matching between registered signals an videofluoroscopy is not correct. For example, in figure 3, the period F2-F1 si simultaneous to period E2-E1. In figure 7, HB2-HB1 happens before TC2-TC1. Table 4 matches HB2-HB1 with E2-E1 and TC2-TC1 with F2-F1.

The sEMG sensors placement was not adequate. If both sEMG electrodes compose a bipolar configuration, the obtained signal is the difference between left and right activation, which is essentially noise.

Also, the air-flow signal was not included in this analysis.

Statistical analysis to verify the algorithm was not clear. Statistical design should be improved.

A comparisson between smokers and non-smokers were made using the non-verified method. There is no reference on the differences between smokers and non-smokers in the intruduction.

There are parts of the method and discussion within the results section. 

The conclusion does not match the results.

I can see the potential of this work, specially regarding the engineering development. But there are some serious design flaws. 

I attach a file with comments.

Author Response

Journal: Sensors (ISSN 1424-8220)

Manuscript ID: sensors-478341

Type: Article

Title: Non-invasive Swallowing Function Measurement: Sensors, Verification, and Clinical Applications

Authors: Wann-Yun Shieh et al. 

Revision Notes

Dear Editors and Reviewers,

    We are submitting the revised manuscript entitled “Non-invasive Swallowing Function Measurement: Sensors, Verification, and Clinical Applications” (ID: sensors-478341) to “Sensors” for replying the first-round reviewing. We have carefully revised the manuscript according to each reviewer’s comments. The major revisions include:

1.     The main goal of this study was revised in Abstract and Section 1. 

2.     Two references about the effect of smoking were added in the reference list and introduced in Section 1.

3.     The detailed sEMG configuration and connection was added in Section 2.1. 

4.     Additional analysis about the coordination of swallowing and respiration was added in Section 2.2 and Section 3.2.2.

5.     The method of Statistical Analysis was added in Section 2.6.

6.     The verification of the algorithm was revised and rewritten in Section 3.1

7.     The conclusion has been rewritten to fit the results. 

Other ambiguous parts were rewritten and new paragraphs and sentences were added when necessary. Below we provide the detailed reply to the comments. 

We greatly appreciate your careful and expeditious review of this manuscript. 

Sincerely yours,

Wann-Yun Shieh, Ph.D.
Associate Professor
Department of Computer Science and Information Engineering,

Chang Gung University

For Reviewer 2

Thank you very much for your kind and great help in reviewing this work. 

Comment 1

The purpose of the study is not clear. Is this a viability study? It is not clear why you decided to include a comparison between smokers and non-smokers.

Reply: 

(1)  The main goal of this study was revised in Abstract and Section 1. Please check Line 22-23, 29-30, 32-36 in Abstract, and Line 87-99 in Section 1 Introduction.

(2)  Two references [17, 18] about the effect of smoking were introduced in Section 1 to make up the motivation of the selected application. Please check Line 91-94 in Section 1 Introduction.

Thanks.

Comment 2:

This study does not verify the method or the algorigthm. Time-period matching between registered signals an videofluoroscopy is not correct. For example, in figure 3, the period F2-F1 si simultaneous to period E2-E1. In figure 7, HB2-HB1 happens before TC2-TC1. Table 4 matches HB2-HB1 with E2-E1 and TC2-TC1 with F2-F1.

Reply: 

Thanks a lot for this important comment. We found that the respiration apnea period (SAD) and the sEMG duration (sEMGD), actually, cannot be compared with VFSS because the respiration events and the submental muscle movement could not be monitored visually by VFSS. Thus we removed the comparison of (HB2-HB1) and (E2-E1) from Table 3 and Table 4. Please check Line 224-238, Table 3, and Table 4 in Section 3.1. In addition, we upgrade Fig. 7 by labelling the positions of the thyroid cartilage and FSR, and added Table 5 to compare the timing difference between F1 and TC1, F2 and TC2, and F3 and TC3. 

Comment 3:

The sEMG sensors placement was not adequate. If both sEMG electrodes compose a bipolar configuration, the obtained signal is the difference between left and right activation, which is essentially noise.

Reply: 

Thanks for this comment. We used a pair of Biopac bipolar electrodes (Biopac EL500, BIOPAC systems Inc., Goleta, CA, USA) to measure sEMG. The Biopac bipolar electrodes provided a differential EMG measurement in the bipolar configuration. Also, the EMG signal was sampled at 1024 Hz and amplified by a factor of 1000 with a differential amplifier and transmitted through a wireless EMG transmitter (Wireless Dynamometry-EMG BioNomadix Transmitter, BIOPAC systems Inc., Goleta, CA, USA). Raw data from the EMG channel is primarily bandlimited from 5.0 Hz to 500 Hz, with sub-filtering options. Thus the noise can be reduced effectively. We have added this description in Section 2.1. Please check Line 106-113 in Section 2.1

Comment 4:

Also, the air-flow signal was not included in this analysis.

Reply:

In Section 2.2, the air-flow signal has been included for measuring the respiration apnea time (A2-A1). In addition, an additional air-flow analysis (respiratory phase patterns pre- and postswallowing) was also included in this study. Please check Line 141-151 in Section 2.2

Thanks.

Comment 5:

Statistical analysis to verify the algorithm was not clear. Statistical design should be improved.

Reply: 

Thanks for this comment. The description about the statistical analysis has been moved to Section 2.6 and improved. Please check Line 214-218.

Comment 6:

A comparison between smokers and non-smokers were made using the non-verified method. There is no reference on the differences between smokers and non-smokers in the introduction.

Reply: 

Thanks for this comment. Two references [17, 18] about the effect of smoking were introduced in Section 1 to make up the motivation of the selected application. Please check Line 91-94 in Section 1 Introduction.

Comment 7:

There are parts of the method and discussion within the results section.

Reply:

Thanks for pointing out this problem. We have moved the original sentences in Line 264 to Section 4 Discussion (Line 324-327). 

Comment 8:

The conclusion does not match the results.

Reply: 

Thanks for pointing out this problem. We have rewritten the conclusion. Please check Line 341-346 in Section 5 Conclusion. 

Other comments:

Reply:

(1) Line 18-19: multiple sensors -> the surface electromyography sensor, the nasal airflow sensor, and the force sensing resistor sensor

(2) Line 58: pasted -> glued

(3) Line 60: midline cervical location -> the midline of the anterior neck

(4) Line 64: Previous researches -> Previous studies

(5) Line 84-85: Signal patterns… -> The coordination of the swallowing and the respiration…

(6) Line 104-105: …without affecting a participant’s swallowing function-> …with the least interference in swallowing.

(7) Lin 165: Table 2, remove the column of “p-value”

(8) Line 212-213: Not knowing the swallowing limit of each 172 participant in advance… -> The maximum swallowing volume was limited at 20 mL according to the previous studies [15,16].

Round 2

Reviewer 1 Report

Actual numbers of swallows in pre- and post-swallowing respiratory phase patterns between the two groups should be presented.

Reviewer 2 Report

Dear authors, 

I have carefully reviewed you manuscript and found some major flaws I would like to addressed.

You applied an independent t test to compare dependent samples. Samples are dependent because you performed the measurements on the same participants (VFSS method and sensor-based method). Then, you compared only FSR with VFSS. You found statistical differences between both methods (Table 4) and concluded (in the same paragraph) that you arrived to the same outcome with both methods. That is not correct.

Your main goal was to compare your method with the VFSS method. It was not achieved. And then, you included a comparison between smokers and non-smokers which was not the aim of your study.

Also, there are drafting flaws. For example: The aim of the study should be within the introduction section (it is only included within the abstract). Results and conclusions should not be within the introduction section (L 87-99).